# Tuning the Extracellular Vesicles Membrane through Fusion for Biomedical Applications

**DOI:** 10.3390/jfb14020117

**Published:** 2023-02-19

**Authors:** Mamata Karmacharya, Sumit Kumar, Yoon-Kyoung Cho

**Affiliations:** 1Center for Soft and Living Matter, Institute for Basic Science (IBS), Ulsan 44919, Republic of Korea; 2Department of Chemical Engineering, Ulsan National Institute of Science and Technology (UNIST), Ulsan 44919, Republic of Korea; 3Department of Biomedical Engineering, Ulsan National Institute of Science and Technology (UNIST), Ulsan 44919, Republic of Korea

**Keywords:** diagnostics, exosome, fusion, liposome

## Abstract

Membrane fusion is one of the key phenomena in the living cell for maintaining the basic function of life. Extracellular vesicles (EVs) have the ability to transfer information between cells through plasma membrane fusion, making them a promising tool in diagnostics and therapeutics. This study explores the potential applications of natural membrane vesicles, EVs, and their fusion with liposomes, EVs, and cells and introduces methodologies for enhancing the fusion process. EVs have a high loading capacity, bio-compatibility, and stability, making them ideal for producing effective drugs and diagnostics. The unique properties of fused EVs and the crucial design and development procedures that are necessary to realize their potential as drug carriers and diagnostic tools are also examined. The promise of EVs in various stages of disease management highlights their potential role in future healthcare.

## 1. Introduction

Extracellular vesicles (EVs) have been demonstrated to serve as key players in intercellular communications in the body through the membrane fusion process and are being recognized as potential circulating biomarkers for many diseases [1,2,3,4,5]. EVs are actively released by all types of cells and can be observed in biofluids. They act as cellular substitutes by transporting proteins, mRNA/miRNA, and DNA from parental cells to other cells [6,7,8,9,10].

The plasma membrane of an EVs plays an important role in defining the closed volume for sustaining intra- and intercellular activities. It not only acts as a border but also mediates the exchange of physical and chemical information between the cell and its environment [11,12,13,14,15,16]. Exosomes are a type of EVs that are formed through the inward folding of endosomal membranes, with an average size of 100 nm [17,18,19]. The creation of exosomes begins with the inward budding of the cellular plasma membrane, forming an endosome. This is followed by the formation of intraluminal vesicles through the further inward budding of the limiting membrane within the endosome, creating a multivesicular body (MVB) [20,21]. Throughout this phenomenon, trans-membrane proteins, peripheral proteins, and cytosolic contents are all integrated into the invaginating membrane [22,23]. These MVBs have the ability to fuse with a cell’s plasma membrane and exocytotically release vesicles into the extracellular environment [24,25].

Membrane fusion is the process whereby two separate plasma membrane vesicles merge and become one; it is essential for communication between membrane-delineated compartments [26,27,28,29,30,31,32,33,34]. The most studied processes involving EVs membrane fusion are endocytosis and exocytosis, whereby an EV’s membrane vesicles fuse with the cell membrane to uptake or release their contents into the intracellular or extracellular environment, respectively [35,36,37,38]. Numerous membrane fusion processes have been presented using a variety of molecular compositions on the plasma membrane surface that tether to or dock with membranes and bring them into close proximity; additionally, they locally disturb the membranes to reduce the energy barriers for fusion [39,40,41,42,43].

Accordingly, the perspective herein considers methods for the fusion of EVs with membrane vesicles (EVs, liposomes, and living cells) to bring EVs in close proximity to other vesicles, along with their corresponding applications in diagnostics and therapeutics (Figure 1). While synthetic vesicles are commonly utilized for delivery purposes after the modification of their membrane, they present several challenges such as a limited half-life, the activation of the reticuloendothelial system for clearance, low biocompatibility, and high immune suppression. Fusion with exosomes offers potential solutions to these challenges, as exosomes contain complex lipid components that provide a favorable environment during interactions [44]. Additionally, the delicate and complex nature of EVs often prohibits the loading of multiple molecules within a single EV. The fusion of EVs with liposomes, exosomes, and cell membranes offers an increased loading capacity, stability, and biocompatibility, with reduced immunogenicity [45,46,47]. While there were previous reviews that briefly mentioned the potential of liposome and exosome fusion as a novel approach for therapeutic applications [48,49,50], no previous review has focused on the EVs fusion strategy and its biomedical applications in diagnostics and therapeutics. This study focuses on bioengineering EVs through membrane fusion strategies. It covers three main areas of interest, which are membrane fusion strategies, examples of vesicles fusing with EVs, and biomedical applications of fused EVs. The review provides valuable insights that could guide the development of innovative strategies for EVs bioengineering and open up new possibilities for future research in this field.

## 2. Strategies of Membrane Fusion

A lipid component plays a crucial role in biological membrane fusion. The mechanical properties of the lipid matrix determine the energy barriers in membrane fusion to a large degree, as they dictate the mechanical properties of the lipid matrix and thereby influence the energies of the intermediates involved in the fusion process. This mechanism depends not solely on the lipid composition but also on external factors, such as pH and temperature [51]. Based on these factors, artificial fusion processes are generally based on pH differences, freeze–thaw cycles, extrusion, polyethylene glycol (PEG), and natural incubation.

### 2.1. pH-Mediated Fusion

The lipid bilayer, a key component of biological membranes, affects membrane fusion through its fluidity, curvature, and charges on the lipid headgroups [52]. The pH level of the environment can also have an effect on the membrane fusion. In an acidic environment, the membrane bending modulus increases [53,54], causing the reorientation of the lipid polar group, which may change the energy profile of lipid membrane fusion [55]. Furthermore, researchers have demonstrated the potential of the lipid bilayer in targeted membrane fusion by engineering the vesicular stomatitis virus G-protein (VSVG) on exosomes and membrane vesicles. Yang et al. harvested VSVG-encoded exosomes from transfected HEK293T cells, which showed exosomal fusion with the targeted cell membrane at low pH values [56]. Similarly, Ren et al. modified the membrane vesicles using the VSVG and N3 group to identify the tumor through membrane fusion [57]. This has shown promise in delivering functional membrane proteins and identifying tumors through membrane fusion.

### 2.2. Freeze–Thaw-Cycle-Mediated Fusion

The freeze–thaw process can have a significant impact on the physical properties of the lipid bilayer, which affect the energy barriers involved in the membrane fusion [58]. During the freezing process, the expansion of water content within the lipid bilayer can lead to mechanical stress and result in changes to the fluidity, curvature, and charges on the lipid headgroups. These changes can then facilitate the interaction between the membranes and promote the fusion process upon thawing [59,60]. Researchers have leveraged this process to fuse giant unilamellar vesicles with small unilamellar vesicles to construct an artificial cell. Using the freeze–thaw method, Akiyoshi and the team created a hybrid exosome by combining the membranes of exosomes obtained from Raw264.7 and CMS7 cancer cells with liposomes [61]. Similarly, Liu and colleagues achieved a 97.4% fused exosome–liposome hybrid after three freeze–thaw cycles [62]. This freeze–thaw method process is simple and quick and avoids contaminating the exosome membranes with unwanted chemicals (such as calcium or PEG) used in other chemical fusion processes [61]. However, repeated freeze–thaw cycles may compromise the membrane’s integrity and destroy the biomolecules contained inside it. To ensure the success of this process, it is critical to carefully consider the number of freeze–thaw cycles used and to monitor the integrity of the lipid bilayer throughout the process.

### 2.3. Extrusion-Mediated Fusion

Extrusion-based membrane fusion is a process that involves the fusion of two lipid membranes through the application of high pressure. This is achieved by bringing the two membranes close together and then applying pressure through an extrusion process using a filter or nanopore. The high pressure causes the lipid bilayers to deform and inter-digitate, resulting in the formation of fusion pores and the eventual fusion of the two membranes [63,64]. This process is highly controlled and efficient and has been widely used in the preparation of liposomes and other lipid-based drug delivery systems [65]. For example, researchers have used extrusion to prepare exosome–liposome hybrid nanoparticles, combining the characteristics of both liposomes and exosomes [66]. However, it is important to note that the high pressure generated during the extrusion may damage the integrity of the exosome membrane.

### 2.4. Polyethylene Glycol-Mediated Fusion

The PEG method for membrane fusion involves using PEG-modified lipids or PEG-conjugated liposomes to promote the fusion of two lipid membranes. PEG reduces the interaction energy between the lipid bilayers and lowers the energy barrier for membrane fusion, making it easier for the two membranes to fuse [67]. This method has been widely used in the preparation of liposomes for drug delivery and the study of membrane fusion [68]. Piffoux et al. added PEG to liposomes to enhance the fusion efficiency, which delivered the PEG molecules onto the engineered exosome surface to lengthen the duration of their circulation [69]. Though it increased the fusion rate and stable activity in physiological conditions, it did not effectively bypass the reticuloendothelial system [70].

### 2.5. Natural Incubation

The natural incubation process for membrane fusion involves the spontaneous fusion of lipid membranes through electrostatic or hydrophobic interactions based on the physicochemical components of mixed vesicles. This process has a low risk of damaging the vesicles and their contents but has a low fusion efficiency [49]. Lin et al. used this process to create an exosome–liposome hybrid for gene therapy, encapsulating the CRISPR/Cas9 expression vector [71].

## 3. EVs and Fusion Membranes

EVs (including exosomes) contain a variety of cellular components, including DNA, RNA, lipids, metabolites, and cytosolic and cell-surface proteins, reflecting the cell of origin [72,73,74,75]. Both trans-membrane and lipid-bound extracellular proteins, such as lactadherin, endosome-associated proteins, and tetraspanins, are present in exosomes [44,76,77]. Tetraspanins (such as CD9, CD63, and CD81), a subfamily of proteins with four transmembrane domains, are particularly abundant in exosomes among the trans-membrane proteins [78,79]. Tetraspanins are used for exosome quantification and characterization because they are highly expressed and also engage in membrane trafficking and biosynthetic maturation [80,81]. By contrast, integrins, selectins, and CD40 ligands are more abundant in microvesicles (average, >100 nm) [82,83,84]. EVs are enriched with particular trans-membrane protein receptors (such as epidermal growth factor (EGF) receptors (EGFRs)) and adhesion proteins (such as epithelial cell adhesion molecules), thereby reflecting their origin from the plasma membranes of cells [85,86]. As many of these trans-membrane proteins are implicated in the pathogenesis of several diseases, they are considered as potential biomarkers. These proteins are responsible for the fusion of the biological membranes of EVs that are essential for the operation of all living organisms, from cell–cell communications to more complex functions [87,88,89]. These biochemical mediating fusions are structurally diverse, follow the merging of two bilayers, and appear as a common pathway involving a sequence of structurally distinct intermediates [90,91,92]. The process begins with loose protein-mediated bilayer membrane contact and progresses to the tight adhesion of the membranes while preserving the integrity of the bilayer [93,94]. In this section, we discuss the artificial fusion of EVs with both synthetic vesicles, such as liposomes, and natural membrane vesicles, such as EVs or cell-derived membranes.

### 3.1. EVs Fusion with Liposomes

Liposomes are synthetic phospholipid vesicles that have potential applications in drug delivery and targeted therapy due to their biocompatibility, biodegradability, and stability. Although they are synthetic, they still have some beneficial properties for use in medical applications. However, they differ from natural membrane vesicles in terms of biocompatibility and bioinertness [95,96,97]. Therefore, the fusion of exosomes with other vesicles, such as liposomes, can change the properties of the fused exosomes, highlighting the importance of understanding the properties of both exosomes and liposomes for medical applications.

Nishio et al. developed a pH-dependent fusion of an exosome membrane with a supported lipid bilayer to control the number of gramicidin A exosomes in the membrane [98]. Reportedly, the exosome membrane fusion assay using HEK-293 and MCF-7 exosomes was improved at a pH of 6.0; the initial rates of membrane fusion for the MCF-7 exosomes were higher than those for the HEK-293 cells. Using the fusion technique, exogenous functional lipids or peptides can be inserted into a membrane. The desired content can be encapsulated by smoothly fusing the synthetic lipid vesicles with the lipid components of the exosome membrane [99]. This fusion can be facilitated by several different approaches, such as chemically triggered, freeze–thaw cycles, and extrusion methods [50,100,101,102,103,104,105].

As noted above, PEG has recently been used to trigger the fusion of EVs with functionalized liposomes [69]. This innovative technique of modifying EVs for drug delivery applications fuses them with liposomes containing both the membrane and soluble cargos. This technique was used to successfully load exogenous hydrophilic or lipophilic chemicals to EVs without altering their natural composition or biological characteristics (Figure 2A) [69]. In comparison to a drug-free or drug-loaded liposome precursor, the hybrid fused EVs increased the cellular transport efficiency of a chemotherapeutic agent by a factor of three to four. The suggested fusion technique allowed for effective EV loading and the pharmaceutical production of EVs with adaptive activities and drug delivery properties.

Akiyoshi and the team developed a novel and facile membrane-engineering strategy to functionalize the exosome surface by direct fusion with liposomes using a freeze–thaw method (Figure 2B) [61]. This fusion process tuned the exosomal immunogenicity and increased the colloidal stability. They created HER2-containing exosomes in cells expressing the tyrosine kinase receptor HER2 and fused them with phospholipid liposomes as a proof-of-concept for creating exosomes using this membrane fusion technique.

Vader and the team suggested an extrusion method for creating EV–liposome hybrid vesicles by combining the advantageous qualities of both liposomes and EVs as siRNA carriers (Figure 2C) [66]. They created semisynthetic hybrid nanoparticles through lipid-film hydration, followed by extrusion. This hybrid system was used for its gene-silencing efficacy and toxicity to multiple cell lines. Finally, they examined whether the functional regeneration characteristics of derived cardiac progenitor cells retained their properties when using the hybrid EVs for functional regenerative properties.

Rayamajhi et al. reported a method for the fusion of exosomes from macrophages and synthetic liposomes to load doxorubicin for tumor-targeted drug delivery (Figure 2D) [106]. The fused exosomes exhibited an increased toxicity to cancer cells and pH-sensitive drug release under acidic conditions, thereby assisting in drug delivery to cancers that thrive in an acidic environment.

The fusion of EVs with liposomes was found to enhance the colloidal stability and reduce the immunogenicity of the membrane without altering its properties. This hybrid fusion approach has been shown to increase the toxicity to tumor cells and improve the drug delivery efficiency for the treatment of tumors using gene silencing techniques. Furthermore, the gene knock-in and knock-out approach could also be utilized to conduct molecular-level studies and treat various genetic disorders.

### 3.2. EVs Fusion with EVs

Exosomes are bilayer-charged membrane nanoparticles that repel one another under physiological conditions and are stable against fusion. Exosomes are used by cellular systems to deliver biological substances to their destinations without leakage to initiate biochemical reactions. This chemical communication typically requires the exosomes to merge with their target membranes to initiate biological processes such as gene expression. Synthetic molecules mimicking this process of tailored exosome fusion have the potential to revolutionize a wide range of technologies, including drug delivery and the creation of artificial biological systems.

Kumar et al. recently developed a supramolecular chemistry-based bridging of two exosomal membranes that led to a controlled fusion of exosomes (Figure 3A) [107]. In this method, MCF-10A human breast epithelial cell-derived exosome membrane proteins (CD9, CD63, or CD81) were modified with a catechol molecule. A droplet-based microfluidic device generated cell-sized droplets. The catechol-modified exosomes responded to metal salt and formed a supramolecular complex between the plasma membranes that ultimately led to exosome fusion. Using this method, they successfully loaded a variety of enzymes (glucose oxidase, horseradish peroxidase, and β-galactosidase) inside the exosomal luminal to perform multienzyme cascade reactions. Further, they utilized this method to install minimal electron transport machinery into the membranes (adenosine triphosphate (ATP) synthase and *bo*_3_ oxidase) of exosomes to produce bioenergy (i.e., ATP). These energy-producing exosomes were utilized to repair diseased tissues. The generation of ATP within diseased tissues could be important in drug delivery for regulating tissue aging and other disease conditions.

For the treatment of various cancers, RNA interference (RNAi) therapeutics for obstructing the programmed death-1 (PD-1) and programmed death-ligand 1 (PD-L1) pathways have gained considerable attention. Liu et al. developed a pH-responsive fusion method for preparing immunoregulatory EVs by fusing M1-macrophage-derived EVs (M1 EV) with vesicular stomatitis virus glycoproteins (VSV-G); subsequently, these were electroporated with anti-PD-L1 siRNA (siPD-L1) (Figure 3B) [108]. From the in vivo studies, the virus-mimic nucleic acid-modified EVs (siRNA@V-M1 EV) could target tumor tissues after being administered to mice with CT26 tumors owing to M1 EV’s inherent ability to home in on tumors. The direct release of siPD-L1 into the cytoplasm and the subsequent robust gene silencing were made possible by the fusion of VSV-G with cells that effectively blocked the PD-L1/PD-1 connection, followed by an increase in the CD8+ T cell population. When this occurred, the M2 tumor-associated macrophages were encouraged to repolarize to M1 macrophages by the M1 EVs and interferon produced by CD8+ T cells. In this tumor model, the combination of inhibiting the PD-L1/PD-1 pathway, restoring T cell recognition, and repolarizing M1 macrophages via multifunctional EVs may produce satisfactory antitumor activity, thus suggesting its potential as a novel method of cancer treatment.

Studies have demonstrated that using EVs for targeted therapy can enhance biocompatibility, stability, and bio-inertness while also improving their homing property to the local environment. This suggests a novel approach to therapy that utilizes EVs and their fusion with either similar or different types of EVs as an alternative to artificial vesicles.

### 3.3. EVs Fusion with Cell-Derived Membranes

As exosomes can facilitate long-distance communication from donor to acceptor cells by transporting biomolecular cargo, they have been extensively investigated as potential therapeutic agents, either by themselves or as vehicles for the delivery of medication payloads [13,48,103,109]. When administered exogenously to mice, injected exosomes are more effective compared with liposomes at entering other cells and can deliver a functional payload with little immune clearance [107].

Recently, researchers have demonstrated that cardiac stem cells fused with platelet vesicles can reach myocardial infarction injuries (Figure 4A) [110]. To deliver their functional payloads, exosomes are thought to go through back-fusion at the MVBs in the recipient cells. The cellular uptake mechanisms of exosomes and their interactions with the plasma membrane of recipient cells are not well-understood, including even the most basic understanding of whether exosome uptake occurs through endocytosis or direct membrane fusion. In one study, researchers investigated a new strategy for directly functionalizing the cellular membrane via exosome fusion (Figure 4B) [56].

Numerous investigations have revealed that the methods of exosome uptake by recipient cells are controlled in various ways, depending on the type of exosome membrane proteins interacting with the membrane receptors of other cells [81,94,111]. Tetraspanins are a type of exosome surface protein. They are thought to be exosome indicators that aid in the attachment of exosomes to recipient cells, thereby promoting exosome uptake [112,113,114]. For instance, exosomes are attached to and taken up by dendritic cells through the actions of CD9 and CD81 [115]. Glebov et al. identified the exosomal surface protein that regulates the Clathrin-independent endocytosis process in cells as flotillin-1, a microdomain of the plasma membrane [116]. Liu et al. demonstrated that EGF is another exosomal surface protein playing a significant part in the uptake procedure via EGFR-mediated endocytosis [117]. Similar to this, Wang et al. showed that Annexin-A2 controls the endocytic cell entrance [118]. Clathrin-dependent endocytosis, lipid raft-mediated endocytosis, phagocytosis, and/or macropinocytosis are other potential mechanisms for exosome uptake. Recently, Nigri et al. discovered that the cell surface glycoprotein and tetraspanin CD9 are crucial markers of the stromal fibroblast-derived ANXA6+ EVs from cancer-associated tissues [119]. The surfaces of the ANXA6+ cancer-associated fibroblasts isolated from patients with pancreatic ductal adenocarcinoma samples were abundant with CD9. These results imply that pancreatic ductal adenocarcinoma progression is facilitated by CD9-mediated stromal cell signaling. In another recent report, platelet membranes were fused with stem cell-derived exosomes to use their ability to target injured endothelia and pro-angiogenic function (Figure 4C) [120]. These EVs retained their pro-angiogenic capability owing to their innate ability to target wounded vasculatures.

**Figure 4 jfb-14-00117-f004:**
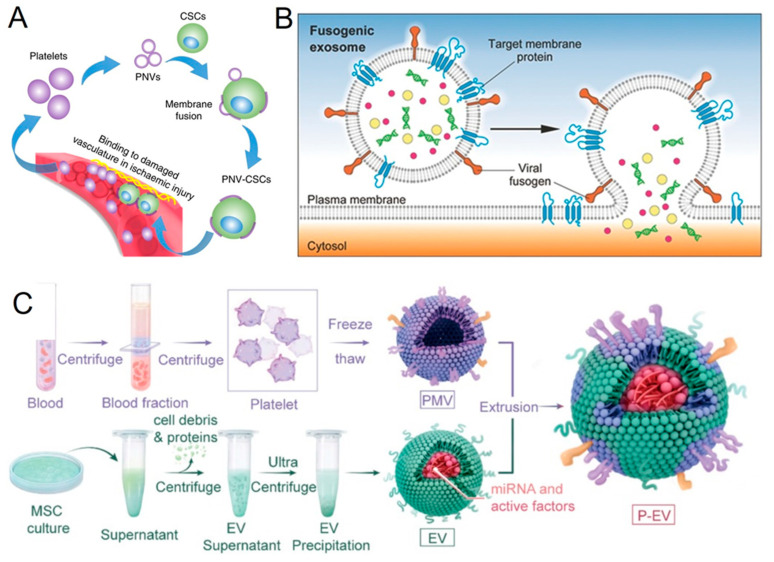
(**A**) Schematic showing the fusion of platelet nanovesicles with the cardiac stem cells to form the platelet nanovesicles-decorated cardiac stem cells. Reproduced from Springer Nature [110]. (**B**) Schematic for the modification of the exosomes membrane by using fusogenic exosomes that fuse with the plasma membrane of cells and functionalize the membrane proteins into the targeted membranes. This method is called “membrane editing”. Reproduced from Willey [56]. (**C**) Schematic presentation shows the preparation of platelet-mimetic EVs by fusing the membranes of EVs with platelet membranes using the extrusion method. Reproduced from Theranostics [120].

## 4. Biomedical Applications of Fused EVs

EVs have the potential to be used in a variety of biomedical applications, including diagnosis and as a vehicle for therapeutic agent delivery [121,122,123]. The biocompatibility, bioinertness, and low immune response of EVs make them a promising option compared to synthetic vesicles [124]. By fusing exosomes with other lipid-based vesicles, the properties of the fused vesicles can be altered to increase the targeting effectiveness and improve the drug delivery [50]. This fusion approach combines the benefits of synthetic and natural vesicles and can be used to design lipid components for the surface of exosomes, creating advanced drug delivery systems.

### 4.1. EVs Fusion for Diagnostic Applications

The precise and accurate identification of target molecules is vital to developing disease-diagnostic devices [125]. To create such platforms, nature is a great place for inspiration, as it has evolved to create extremely sensitive and specific sensing and signaling processes using refined components made up of only a few molecular building blocks [114,126,127,128]. Membrane fusion comprises highly selective molecular recognition mechanisms and can be used for biosensor development; this approach has enormous potential, as it is accompanied by the engagement of a large payload of signal-generating molecules [129]. Ning et al. developed a method for fusing exosomes with liposomes containing reagents for reverse transcriptase, recombinase polymerase amplification, and clustered regularly interspaced short palindromic repeat (CRISPR)-Cas12a (Figure 5A) [130]. For the clinical diagnosis, exosomes were directly captured from plasma through the binding of an antibody (CD81) on the surface and were detected using an enzyme-linked immunosorbent assay. After the fusion of the exosomes and liposomes, guide RNA directed the CRISPR-Cas12a binding to an RT-RPA amplicon, where a quenched oligonucleotide probe was cleaved. The results demonstrated the ultrasensitive detection of SARS-CoV-2 RNA.

Stevens and the team demonstrated a highly specific detection of microRNA via sequence-specific DNA-mediated liposome fusion [131]. Using a common laboratory microplate reader, miR-29a, a well-known flu biomarker, could be detected at levels as low as 18 nM in less than 30 min with good specificity.

Gao et al. described a virus-like fusogenic vesicle (Vir-FV) for enabling the high-throughput, quick, and effective detection of exosomal miRNAs within 2 h [132]. To effectively fuse the Vir-FVs and exosomes, fusogenic proteins on the Vir-FVs can selectively target sialic-acid-containing receptors on the exosomes. The molecular beacons contained in the Vir-FVs specifically hybridize with the target miRNAs in the exosomes upon vesicle content mixing, thus producing fluorescence. By detecting tumor-related miRNAs, the Vir-FVs can be used to distinguish tumor exosomes from normal exosomes when used in conjunction with flow cytometry. This opens the door to the quick and effective identification of exosomal miRNAs for disease diagnosis and prognosis prediction.

The fusion strategies discussed in these studies showcase the potential of using nature as an inspiration for developing diagnostic devices for diseases. Membrane fusion, particularly of exosomes and liposomes, is used to create biosensors with a high sensitivity and specificity for detecting RNA and microRNA. Additionally, virus-like fusogenic vesicles are highlighted as a promising method for detecting exosomal miRNAs for disease diagnosis and prognosis prediction. These advancements hold great promise in the field of disease diagnosis and underscore the importance of interdisciplinary research on connecting biological processes with technology.

### 4.2. EVs Fusion for Therapeutic Applications

Exosomes may have a greater therapeutic impact when fused with liposomes, as the latter can improve their targeting effectiveness [50]. The appropriate exogenous functional lipids or peptides may be injected into the exosome membrane using this technology; correspondingly, therapeutic or imaging materials can be encapsulated within exosomes more effectively and reproducibly [133,134].

One study demonstrated that fused exosomes have an enhanced cellular uptake and are an effective carrier of exogenous hydrophobic lipids [61]. Evers et al. showed the loading of siRNA inside a fused SKOV3 exosome–liposome hybrid and successfully delivered siRNA to numerous cell types [66]. In terms of cellular absorption, toxicity, and gene silencing efficacy, hybrids act functionally differently from liposomes; the behavior varies depending on the recipient cell type. In addition, hybrid vesicles created using exosomes obtained from cardiac progenitor cells (CPC) maintain the functional qualities associated with CPC exosomes, such as the ability to migrate and activate endothelial signaling. In comparison to liposomes alone, CPC EV–liposome hybrid particles facilitate wound healing and stimulate Akt phosphorylation in a dose-dependent manner, thus indicating that the fusion process has no effect on the functional characteristics of the exosomes.

In another study, Tareste and the team utilized simple co-incubation techniques; PEGylated liposomes were fused with exosomes generated from mesenchymal stem cells or human umbilical vein endothelial cells [69]. The fused exosome–liposome hybrid exhibited less macrophage cellular uptake compared with the individuals [135].

To operate as a mechanism for cargo loading, the liposome fusion with exosomes can also be used to deliver hydrophilic or lipophilic molecules to the insides of the exosomes. Whereas Evers et al. showed the loading of siRNAs into CPC–EV hybrids, Piffoux et al. discovered that mTHPC, a tiny anti-tumor photosensitizer loaded into the liposomes, could be successfully encapsulated inside the exosome through membrane fusion [66,69].

In a recent report, Tang et al. reported a membrane fusion technique involving the utilization of the cell plasma membrane as a natural biomaterial alternative to synthetic liposomes (Figure 5B) [110]. Natural cell membranes have built-in targeting capabilities owing to the native proteins in the membranes. The fusion modification of cardiac stem cells with platelet membrane nanovesicles (which have a natural targeting affinity to infarcted heart regions) enabled the realization of the functional features of the cell membranes. After being decorated with platelet nanovesicles utilizing PEG-mediated fusion, the cardiac stem cells, which ordinarily have poor innate homing properties to injury sites, demonstrated dramatically enhanced targeting and retention in an infarcted heart.

In a recent study, Zhang et al. applied cell membrane fusion to stem cell-derived extracellular vesicles known to be effective in heart repair and regeneration post-infarction to improve the delivery efficiency of exosomes to an injured myocardium in a murine myocardial ischemia-reperfusion injury model [136]. In their method, serial co-extrusion was employed to fuse monocyte membranes and bone marrow mesenchymal stem cell-derived exosomes. Despite being functionally angiogenic, stem cells exhibit poor targeting properties that create difficulty in providing therapeutic benefits. By contrast, monocytes have an abundance of adhesion proteins (such as 41, ALB2, and P-selectin glycoprotein ligand I) that promote homing and retention in damaged cardiac regions. After the fusion of mesenchymal stem cell-derived exosomes with the monocyte membrane vesicles, their hybrid membrane vesicle exhibited enhanced targeting to the damaged myocardium and greater cardiac recovery by enhancing endothelial maturation and controlling inflammatory responses.

Paul and the colleague developed a membrane fusion technique using an artificial extracellular vesicle [137]. To remove intracellular contents, they continuously centrifuged human adipocyte stem cells using a succession of filters. The EVs were produced naturally by dividing the parent membrane. The exosomes prepared using this technology exhibited great stability over a 3 w period and strong target drug delivery capabilities without any significant cytotoxicity.

However, one of the major challenges for exosome-based clinical translation is the insufficient number of secreted exosomes. To solve this problem, Jhan et al. developed a method for the mass production of engineered exosomes, fusing exosomes with lipid-based materials (DOTAP, POPC, DPPC, and POPG) using an extrusion technique [65]. Uniform lamellar vesicles with a regulated size of approximately 100 nm were produced, thus enabling a 6- to 43-fold increase in the number of vesicles after isolation. Their findings demonstrated that the lipid extrusion could modify the surface structure and functionality of exosomes by the exogenous loading of siRNA into the exosomes with an approximately 15–20% encapsulation efficiency, thus enabling their mass production while preserving their targeting ability (e.g., a 14-fold higher cellular uptake in lung cancer cells (A549)). Additionally, they achieved an effective gene silencing effect comparable to that of the commercial Lipofectamine RNAiMax.

Lipids can be used to directly label cell membranes to create vesicles resembling exosomes. Wan et al. developed a method for expressing membrane-bound targeting ligands on the surfaces of exosomes by fusion with targeting liposomes via mechanical extrusion [138]. Using this strategy, they conjugated the nucleolin-targeting aptamer AS1411 with Cholesterol poly (ethylene glycol). The conjugate was immobilized on a mouse dendritic cell membrane. The cells were extruded through two filters with hole diameters of 10 and 5 µm to create vesicles resembling exosomes.

Paclitaxel could be loaded into the exosomes and administered in vivo using ultrasound to treat cancer. These findings suggest that extruded cells provide a quick, easy, and affordable method for producing sufficient drug delivery systems incorporating ligands.

The efficiency of fused EVs has been investigated for various biomedical and therapeutic applications, but there are some limitations to their widespread use. These include a lack of standardization in production and testing, a complex and time-consuming regulatory approval process, and safety concerns about long-term toxicity. The high cost and time required for the purification of high-quality fused EVs also limits their accessibility. Although liposome-mediated EV modification has shown some promise, standardization remains a problem. This is because the fusion efficiency depends on the origin of the exosomes and the composition of the liposomes. Regulating the degree of fusion may be challenging, owing to the numerous types of lipids present in synthetic liposomes and the wide variety of exosome membrane proteins. In addition, their properties, such as their cellular uptake, stability, and targeting in the tissue, are dependent on the exosome–liposome ratio, which may render this engineering procedure somewhat unpredictable, thus necessitating the specific regulation of each fusion event [107]. To overcome these limitations and make this technology more accessible, new methods are needed to generate and use fused EVs, and these will require extensive clinical trials before they can be used in humans.

## 5. Conclusions and Future Perspectives

Exosomes, small membrane vesicles released by cells, have emerged as a promising platform for disease diagnosis and therapeutic delivery. The lipid engineering of exosomes, including the addition of targeting ligands, stimuli-responsive components, and immune-evasive components, has improved their potential for drug administration. In recent years, the field of exosome fusion research has seen significant progress, offering exciting possibilities for the improvement of disease diagnosis and therapeutic molecule delivery.

One of the main advantages of the exosome fusion strategy is that it allows researchers to encapsulate a controlled number of molecules in separate vesicles and then combine them into one vesicle during fusion. This improves the physical and chemical properties of the fused vesicles, leading to increased biocompatibility and reduced immune clearance. The fusion process also enhances the stability and loading rate of the vesicles and enables surface modification, making it a promising platform for drug delivery, biologics, and other therapeutic applications.

Although there are some limitations to the use of the fusion strategy, such as a lack of standardization in production and characterization, researchers are continuously working to overcome these limitations and develop a more ideal bio-material. Utilizing a combination of synthetic and natural vesicles can improve the targeting effectiveness of exosomes for diagnostic and therapeutic purposes and provide a strategy for loading materials inside exosomes for improved drug delivery and treatment outcomes.

Several effective fusion techniques, such as PEG-based fusion, mechanical extrusion, pH-based fusion, and molecular bridging, have been developed and are continuously being improved. In the future, the integration of these methods using microfluidic devices, automation, and high-throughput analyses is expected to become more prevalent.

While there are still some production and safety issues to be considered in exosome fusion approaches, the development of clinical-grade exosome fusion methods and their subsequent translation to the clinical setting are continuously being advanced. With a deeper understanding of the issues at hand and continuous efforts to overcome the limitations, the development of off-the-shelf or streamlined exosome-based disease therapies is expected to be accelerated, leading to greater utility and commercial success.

In conclusion, the exosome fusion strategy offers a unique opportunity to improve the physical and chemical properties of vesicles for use in biomedical and therapeutic applications. While there are still some limitations to be addressed, the continued advancement of this field holds great promise for the diagnosis and treatment of various diseases.

## Figures and Tables

**Figure 1 jfb-14-00117-f001:**
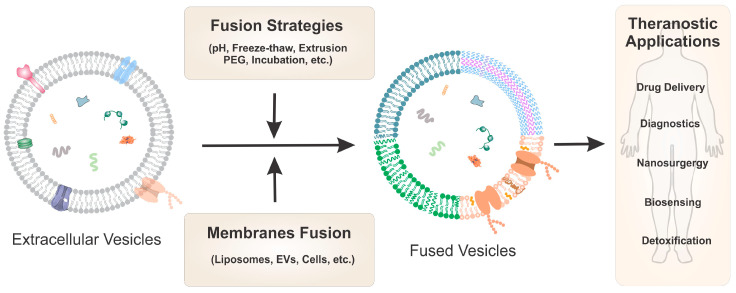
Extracellular vesicles (EVs) are being explored for their potential in precision theranostic applications through the creation of fused systems. Various fusion methods exist, including pH-mediated, freeze–thaw, extrusion, polyethylene glycol (PEG)-induced, and natural incubation. These methods lead to various levels of fusion yield. EVs that are engineered to carry diagnostic molecules, therapeutic agents, or other functional proteins can be further modified on their membrane surface to enhance their targeting capabilities for theranostic applications.

**Figure 2 jfb-14-00117-f002:**
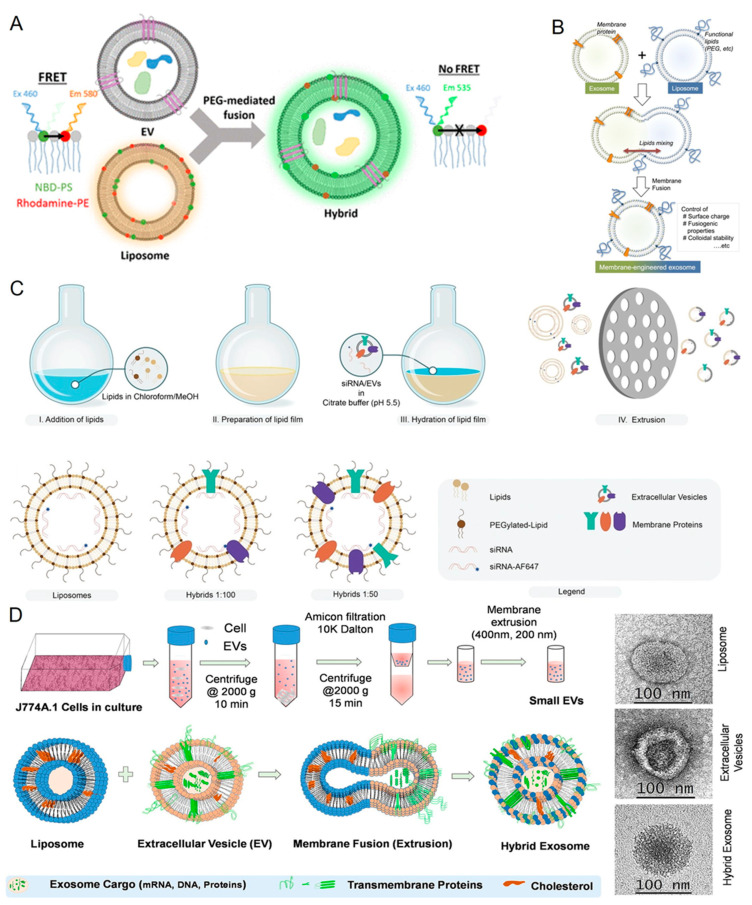
(**A**) EVs are modified by being fused with liposomes to create customized biologic drug delivery systems. PEG-mediated rapid mixing of lipids of EVs and liposomes and fluorescence resonance energy transfer (FRET)-based quantification of membrane mixing during fusion. Reproduced from ACS [69]. (**B**) Schematic for the engineering of exosome–liposome fusion. Genetically modified cells-derived exosomes are fused with PEG-DSPE liposomes. Reproduced from Springer Nature [61]. (**C**) Schematic for the fusion of exosomes with liposomes using thin-film hydration and extrusion to mix the fluorescent and nonfluorescent siRNA. Reproduced from Willey [66]. (**D**) Schematic showing the fusion of immune cell-derived small EVs with synthetic liposomes using the membrane extrusion method. Reproduced from Science Direct [106].

**Figure 3 jfb-14-00117-f003:**
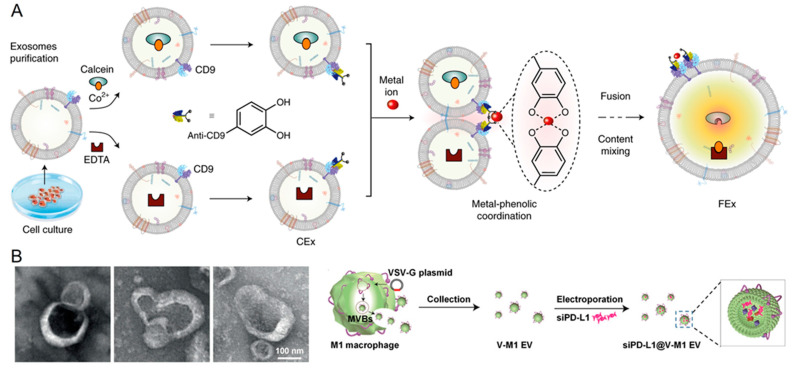
(**A**) Schematic showing the metal-triggered exosome fusion by making the supramolecular complex with the catechol-modified plasma membrane that brings the exosomes together. During this process, multiple reactants were encapsulated together and mixed inside the fused exosomes. Reproduced from Springer Nature [107]. (**B**) Transmission electron microscopic images and schematic presentation of the loading of siRNA into the EVs during fusion. Reproduced from Willey [108].

**Figure 5 jfb-14-00117-f005:**
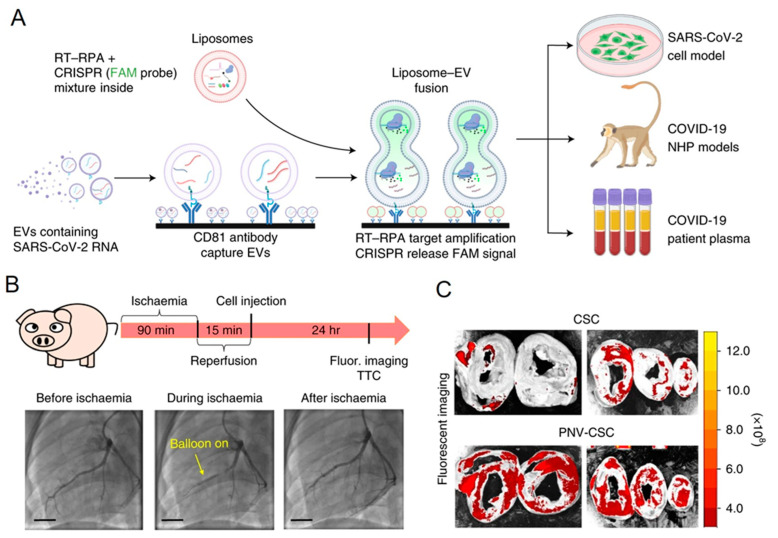
(**A**) Detection of SARS-CoV-2 RNA-positive EVs in plasma using the fusion method with liposomes. Reverse transcriptase–recombinase polymerase amplification-clustered regularly interspaced short palindromic repeat (RT-RPA-CRISPR)-loaded liposomes fuse with plasma EVs in the proposed assay’s schematic, which also shows target amplification by RT-RPA and signal production by the CRISPR-mediated cleavage of a quenched fluorescent probe in proportion to the target amplicon concentration. Cell culture media and plasma from COVID-19 patients and nonhuman primate (NHP) illness models serve as analysis sample types. Reproduced from Springer Nature [130]. (**B**,**C**) Heart injury repair by utilizing stem cells fused with platelet nanovesicles. This schematic shows the pig study design and the angiograms for coronary flow during the placement of a balloon before and after ischemia. Scale bar, 15 mm. Fluorescence imaging of ischemia/reperfusion pig hearts after the injection of platelet nanovesicles-decorated cardiac stem cells. Reproduced from Springer Nature [110].

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
