# Peer review of "Tuning the Extracellular Vesicles Membrane through Fusion for Biomedical Applications"

_jfb, 2023, doi:10.3390/jfb14020117_

Round 1
Reviewer 1 Report
It is a very interesting review article, but should be thoroughly checked by a native English speaker. Some sentences are hard to understand.
You should also follow carrefully the Instructions for authors. In References section you should add doi code of the articles。
These are my supplementary comments on jfb 2184645 manuscript
1. It is an interesting review discussing about the potential applications of EVs fusion with the liposomes, EVs and cells that could be utilized for diagnostics and therapeutics.
2. I consider the topic original and relevant for the field. The specific gap in the field is to improve the diagnostic and therapeutic tools in many diseases, using biomarkers expressed by Extracellular vesicles (EVs)- see references 11-16.
3. The review is based on most recent literature, from very relevant journals, mainly from Nature group, on the subject of bioengineering of Extracellular vesicles (EVs) and exosomes for diagnostic and therapy of different diseases
4. Firstly the authors must improve English language, to be more comprehensive. The discussion could be also improved, to review the benefits, novelty, and limitations of bioengineering of Extracellular vesicles.
5. The conclusions are consistent with the evidence and arguments presented, but could be more synthetic, should not include references
6. The references are appropriate, including up to date literature
7. Figures are very suggestive, reproduced from very relevant literature
Respectfully,
Reviewer 2 Report
The authors have proposed a review article called “Tuning extracellular vesicles membrane through fusion for bio-2 medical application”. They covered different fusion models and demonstrated the potential applications of membrane fusion in diagnosis and therapeutics. A few corrections are needed before further processing.
1. In section 2, a section about pH fusion, freeze-thaw cycle, PEG, extrusion, and other methods before introducing different fusion models can be helpful. They could be classified based on their principle, passive physical fusion, chemical-based, and engineered methods. Especially the engineered method; adding some descriptions of engineering methods will be helpful. The advantages and disadvantages of different methods can be discussed briefly. In lines 249-251, the authors discussed the freeze-thaw cycle advantages, and it would be better to add these discussions in section 2.
2. Instead of describing the advantages of the methods, in the therapeutic application section, focusing on how membrane fusion improves efficacy is essential. For example, membrane fusion enhances encapsulation efficiency and targeting; increases the stability of exosomes; increases the number of exosomes, and stabilizes the release. Examples could be categorized based on their function, if possible.
3. Adding potential side effects, if possible, when discussing the application of some materials in the drug delivery field. And how will we address the problems in the future?
Reviewer 3 Report
1. As your abstract's final sentence, include a "take-home" message.
2. Keywords should be reordered based on alphabetical order.
3. Nothing truly unique in its current state. Because of the lack of novel, the current perspective looks to be a replication or modified literature. The authors must describe their novel in detail. This work should be rejected owing to a major issue.
4. In order to highlight the gaps in the literature that the most recent literature aims to fill, it is crucial to review the benefits, novelty, and limitations of earlier literature in the introduction.
5. The authors usually use “we” in the manuscript. Please change it into passive.
6. Please more highlight biocompability and bioinert aspect in biomedical application for improve the present manuscript quality. Relevan reference from MDPI needs to be adopted as follows:
7. The Effect of Bottom Profile Dimples on the Femoral Head on Wear in Metal-on-Metal Total Hip Arthroplasty. J. Funct. Biomater. 2021, 12, 38. https://doi.org/10.3390/jfb12020038
8. The authors need to improve the discussion in the present article become more comprehensive. The present form was insufficient.
9. Before moving on to the conclusion section, the present perspective limitation must be added at end of the discussion section.
10. Mention further research in the conclusion section.
11. The reference is recommended to be enriched with literature from five years ago. MDPI reference is strongly recommended.
12. The manuscript needs to be proofread by the authors since it has grammatical and language issues.
13. Following the revision step, the authors must provide a graphical abstract.
Round 2
Reviewer 3 Report
Reviewers greatly appreciate the efforts that have been made by the author to improve the quality of their articles after peer review. I reread the author's manuscript and further reviewed the changes made along with the responses from previous reviewers' comments. Unfortunately, the authors failed to make some of the substantial improvements they should have made making this article not of decent quality with biased, not cutting-edge updates on the research topic outlined. In addition, the author also failed to address the previous reviewer's comments, especially on comments number 3 (nothing really novel with cutting edge insight), 4 (not perfectly capture state of the art), and 7 (poor discussion, not something comprehensive). Thank you very much for the opportunity to read the author's current work.
Author Response
We would like to thank the reviewer for taking the time to evaluate our manuscript and provide valuable comments and suggestions which have significantly improved our manuscript. Please see the attachment.
